# Nondestructive Evaluation of Apple Fruit Quality by Frequency-Domain Diffuse Reflectance Spectroscopy: Variations in Apple Skin and Flesh

**Nan-Yu Cheng, Chien-Chih Chen, Bo-Jian Liang and Sheng-Hao Tseng \***

Department of Photonics, National Cheng Kung University, Tainan 701, Taiwan;
nycheng52@gmail.com (N.-Y.C.); jason75122345@gmail.com (C.-C.C.); open1991525@gmail.com (B.-J.L.)
\* Correspondence: stseng@mail.ncku.edu.tw



**Featured Application: DRS techniques might be practical tools for grading fruits or for monitoring fruit changes during long-term storage.**

**Abstract:** The optical properties of fruits, such as light absorption and scattering characteristics, change with biochemical activities during storage. Diffuse reflectance spectroscopy (DRS) systems have been widely applied for noninvasively observing biological tissues. In this study, we used a frequency-domain DRS system to measure the optical properties of apples. Results showed that variations in the chlorophyll, water, and flesh-texture of apples could be noninvasively monitored over time. We also observed substantial differences in the absorption and reduced scattering coefficients between injured and normal apples. The DRS techniques could be used for apple grading, and, by extension, for monitoring the quality of other fruits.

**Keywords:** nondestructive; fruit; apples; optical properties; spectroscopy

---

## 1. Introduction

Nondestructive assessment of the quality of fruits and vegetables is important for production and sales. Apples have been the subject of many studies, because they are one of the most economically important fruits in the world. At present, the quality of apples can be qualitatively evaluated by observing their size, color, and shape, or quantitatively determined by chemical or optical methods. The charge-coupled device (CCD) image analysis, hyperspectral image analysis, and multispectral imaging with computer vision algorithms are commonly used for apple sorting and grading [1–3]. With advances in measurement systems and algorithms, the recognition rate of existing methods for grading healthy and defective apples is up to 90% [4]. However, these techniques are limited to obtaining information on the surface features of apples. Thus, imaging-based systems cannot easily detect damaged apples, especially those with recent bruises with a color similar to that of healthy apples [5].

The changes in the optical absorption and scattering in apple flesh need to be evaluated to monitor the interior condition of the apples. The optical properties of tissues could be indicative of their chemical and physical properties. Numerous studies on measuring the optical properties of fruits using advanced optical techniques that are mainly based on the radiative transfer theory have been conducted [6–8]. These optical methods have been widely applied for the noninvasive characterization of tissue properties through measurements of light–tissue interaction phenomena such as reflectance, propagation, and attenuation [7,9]. Among the various optical methods, diffuse reflectance spectroscopy (DRS) is useful for monitoring apple quality. For example, Cubeddu et al.

used a femtosecond pulse laser source, and measured the diffuse reflectance that indicates the pulse broadening effect caused by apples, to derive the absorption and scattering properties of the fruits [10]. Saeys et al. estimated the optical properties of apple skin and flesh in the wavelength range of 350–2200 nm [11]. Anderson et al. measured the absorption and scattering properties of apples using the spatial frequency-domain imaging to determine the difference between bruised and nonbruised apples [12]. However, the change in the optical properties of apple skin and flesh during ripening and postharvest storage has not been reported. In addition, no research has established the relationship between the sampling depth and optical properties at different parts of apples. Furthermore, there is no research exploring the difference in the optical properties between normal apples and injured apples over time.

Photon absorption and scattering can be characterized by the absorption coefficient ($\mu_a$) and reduced scattering coefficient ($\mu_s'$), respectively. The absorption coefficient is related to the fruits' chemical components, including chlorophyll and water, and the absorption spectrums of chlorophyll-a and water manifest their absorption peaks at approximately 660 and 970 nm, respectively [13,14]. The reduced scattering coefficient is related to the physical structure of fruits such as cell structure. Chlorophyll degradation mainly occurs during leaf senescence or fruit ripening [13]. The cell wall depolymerization and the dissolution of the middle lamella lead to cell separation during storage [15]. The optical properties are potentially useful for estimating the quality, freshness, and firmness of fruits through the detection of the differences in absorption and scattering based on the two natural changes. Light absorption is related to tissue chemical constituents, such as moisture content and fruit ripeness, while light scattering is influenced by physical and structural properties, such as firmness and cell size [8]. In addition, variations in the source–detector separation (SDS) settings would result in variations in sampling volumes and depths. Specifically, short and long SDSs are useful for studying apple skin and outer flesh, respectively. We aim to apply our frequency-domain DRS system at different SDSs to noninvasively monitor the optical properties of apple tissues, from the skin to the outer flesh, to provide information on apple fruits, and to detect damages that are obscured by the apple skin.

## 2. Materials and Methods

### 2.1. Apple Samples

The commercial apple cultivar, Gala, from the United States was used. Those were purchased from a local market. The study comprises two experiments conducted at different times. Because of a prolonged time needed for the measurement by FDPM system, only six apple samples was used for each experiment. The apples in the first experiment were kept in an air-conditioned room, and the temperature and relative humidity were set to 25 °C and 60%, respectively. The apples in the second experiment were stored in the refrigerator, with a temperature and relative humidity of 4 °C and 30%, respectively. The apples in the refrigerator were kept in plastic bags to reduce water evaporation. The DRS measurements of the optical properties of each apple sample were performed 1, 7, and 14 days after purchase.

### 2.2. Frequency-Domain Photon Migration (FDPM) Diffuse Reflectance Spectroscopy (DRS) System and Artificial Neural Network (ANN)

The configuration of our FDPM DRS system is shown in Figure 1. Two laser diodes with wavelengths of 660 and 980 nm, corresponding to the chlorophyll-a and water absorption peaks, were used in the system. They were installed on laser diode mounts (LDM9T, Thorlabs, Newton, NJ, USA), and driven by RF signals generated from a network analyzer (N5230C, Agilent, Santa Clara, CA, USA) and a DC source (LDC-3908, ILX, St. Charleston, SC, USA) to produce an intensity-modulated light at 10 dBm sinusoidal RF power in the range of 10–500 MHz. The network analyzer swept through 201 frequency points that were equally spaced between the starting and stopping frequencies. The RF switch was programmed to direct the RF signal to one diode laser at a time. The diffuse

reflectance of apples measured at a certain SDS was coupled to an avalanche photo detector (APD, C5658, Hamamatsu, Japan) to convert the RF signal feeding back to the network analyzer's receiver port. The amplitude data represent the ratio of the sample to reference signal amplitudes, while the phase data represent the phase difference between the sample and reference signals.

The source and detector optical fibers used in the system were two-step index fibers with 440 μm core size and NA of 0.37. The measured variables for the FDPM measurements include the amplitude and phase of the intensity-modulated light. In order to separate amplitude and phase changes due to the instrument response, the system must first be calibrated with a standard of known optical properties. Calibration measurements are performed on homogeneous silicon or liquid phantoms of known properties in order to determine the instrument response of the system. We measured 5 points at the relatively flat spot of each apple, and each point was close by. The horizontal translation of the source and detector fibers was precisely controlled by a translation stage of accuracy within 10 μm. The SDS was set to 1 and 5 mm, to investigate the relatively superficial and deep regions of the apple, respectively. We used a second translation stage to provide the vertical translation of the source and detector fibers. This vertical translation of the fibers was initiated before the measurement point was changed, to ensure that the fibers would not scratch the apple skin during measurement point adjustment. A vertical translation was initiated to return the fibers to their original measurement position after measurement point adjustment.

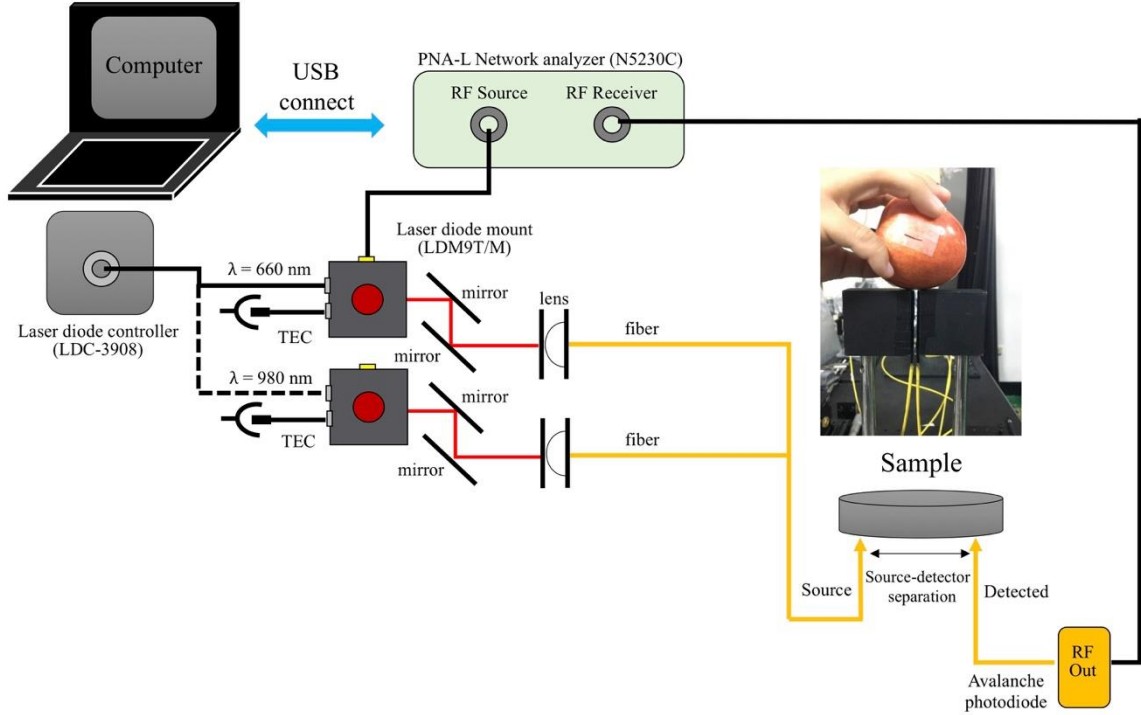

**Figure 1.** Schematic of the frequency-domain diffuse reflectance spectroscopy (DRS) system configuration.

### 2.3. Optical Properties Measurement

Typical diffuse reflectance spectroscopy (DRS) systems can work with photon transport models to determine the absorption coefficient ($\mu_a$) and reduced scattering coefficient ($\mu_s'$) of tissues. The $\mu_a$ and $\mu_s'$ are calculated by the phase and amplitude of photon density waves which are derived from the diffusion equation with extrapolated boundary condition [16]. In the frequency-domain method, the source is sinusoidally modulated at frequency *f*. The physically measured quantities in

a frequency-domain DRS measurement are the phase change and the amplitude modulation of the light that has propagated through the tissue, which are given by:

$$Amp(\rho, \omega) = \left\{ \mathrm{Re}[R(\rho, \omega)]^2 + \mathrm{Im}[R(\rho, \omega)]^2 \right\}^{1/2} \tag{1}$$

$$Phase(\rho, \omega) = \tan^{-1} \frac{\mathrm{Im}[R(\rho, \omega)]}{\mathrm{Re}[R(\rho, \omega)]} \tag{2}$$

Here, $\omega = 2\pi f$, $\rho = \left(x^2 + y^2\right)^{1/2}$ and $R(\rho, \omega)$ is the spatially resolved reflectance.

However, at short SDS, as is essential for investigating superficial tissue volumes, most of the photon transport models are invalid. We constructed a database for the ANN training and calculated the diffuse reflectance for various optical properties and various modulation frequencies. We defined the ranges of $\mu_a$ and $\mu_s'$ to be from 0.001 and 0.1 mm$^{-1}$, and from 0.5 and 5 mm$^{-1}$, respectively. Other simulation parameters are listed below: the refractive index of sample was 1.33 for apple studies, the anisotropy factor was 0.9, and the phase function was the Henyey–Greenstein phase function. Thus, we used a Monte Carlo-based ANN to determine the sample optical properties from the frequency-domain diffuse reflectance collected at short SDS. We have carefully verified the ANN in our previous study [17].

The source of the optical properties and the procedure for determining it is illustrated in Figure 2. The intensity-modulated light was launched into the tissue, thereby resulting in the propagation of diffuse photon density waves. Amplitude demodulation and phase delay detected by the network analyzer were compared with the diffusion model. Then, a Levenberg–Marquardt minimization algorithm was used to calculate the absorption and reduced scattering coefficients. The photon transport model assumes that the sample has a flat surface, and the source and detector fibers are flush with the sample surface. We found that 5 mm was the longest SDS at which the validity of this assumption may be confirmed for measuring apples. Thus, the longest and shortest SDSs were set to 5 and 1 mm, respectively. On all the measurement days, we collected 5 frequency-domain reflectance values at SDSs of 1 and 5 mm on a certain spot of each apple, and the source and detector fibers were replaced on the apple surface after each frequency-domain reflectance was collected. The $\mu_a$ and $\mu_s'$ were calculated from 5 reflectance values at 5 points separately by diffusion equation or ANN model. After that, the $\mu_a$ and $\mu_s'$ were averaged and then regarded as the optical parameters of each apple.

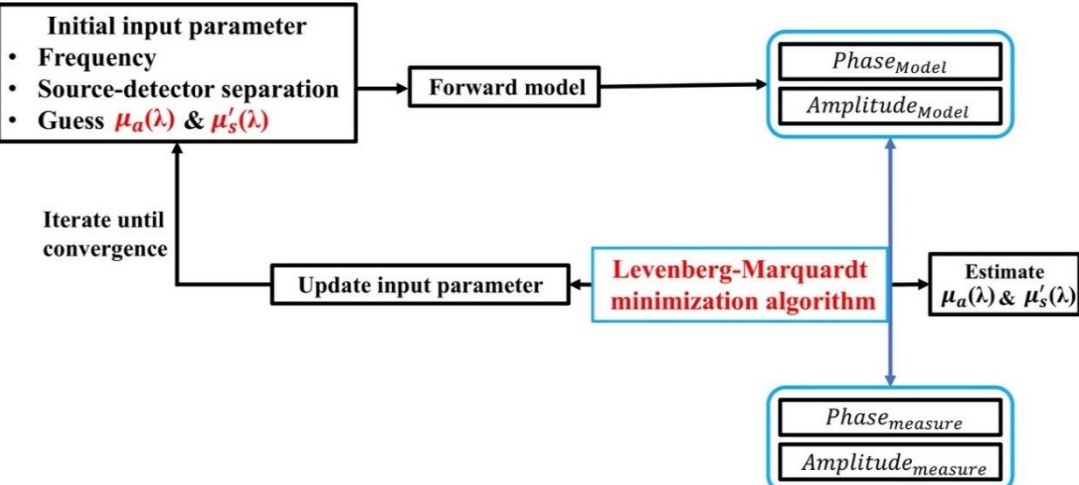

**Figure 2.** Flowchart of the measurement of optical properties.

## 3. Results and Discussion

### 3.1. Optical Properties of Gala Apples at Different Depths

The optical properties of the apples recovered by our frequency-domain DRS system at SDSs of 1 and 5 mm are displayed in Figure 3. The error bars indicating the standard deviation of five measurements for an apple are presented in the plots. Considering the computational error of ANN model and system stability, the maximal errors of $\mu_a$ and $\mu_s'$ recovered by our FDPM system were less than 10%. The impact of the errors that came from the system were relatively less compared to the impact of the deviation made from the individual character of each apple. At 1 mm SDS, our frequency-domain DRS system was sensitive to the property variation in the apple skin, whereas at 5 mm SDS, the system's probing volume mainly accounted for the apple outer flesh. The $\mu_a$ recovered at 660 and 980 nm represented the absorption of chlorophyll and water in the apple tissues, respectively. The $\mu_s'$ is related to the structural characteristics of apples; hence, it is potentially useful for estimating fruit firmness [18]. In Figure 3a, the $\mu_a$ at 660 nm is much higher than that at 980 nm when measuring the optical properties at 1 mm SDS. This demonstrates that the absorption of chlorophyll was more significant than the absorption of water in the apple skin. In contrast to the optical properties recovered when the SDS is 5 mm, the $\mu_a$ at 980 nm is higher than that at 660 nm. This demonstrates that the absorption of water was more significant than that of the chlorophyll in apple outer flesh. In Figure 3b, the $\mu_s'$ at both wavelengths of 660 nm and 980 nm and SDS of 1 mm exceeds those at 5 mm, thus demonstrating that the firmness of apple skin is greater than the firmness of its outer flesh. Moreover, the error bar of the $\mu_a$ is large at 660 nm at 1 mm; therefore, we speculate that the distribution of chlorophyll is uneven in apple skin. The optical parameters recovered at SDS 1 mm have contribution mainly from skin, and those recovered at 5 mm have contribution not only from flesh but also from skin, but flesh is more predominant than skin.

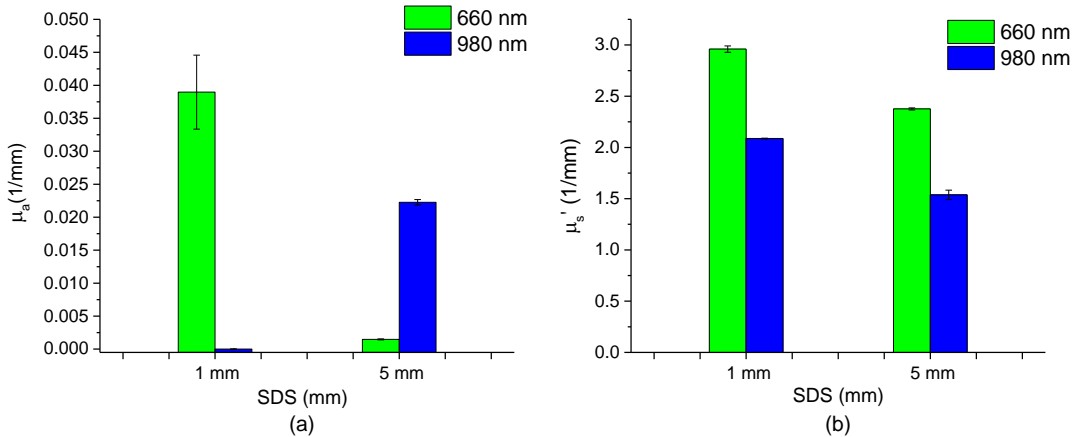

**Figure 3.** (**a**) Absorption and (**b**) reduced scattering coefficients of an apple recovered at 660 nm wavelength (green bar) and 980 nm (blue bar) at source–detector separation (SDS) of 1 mm and 5 mm.

### 3.2. Optical Properties of Gala Apples at Different Depths and Time During Storage

We measured the absorption and reduced scattering coefficients at two wavelengths and two SDSs of six apples. Figure 4 shows the average amplitude and phase of all apples at 660 and 980 nm. The frequency range employed in the fitting was from 10 to 500 MHz because the data collected at modulation frequencies higher than 500 MHz were noisy and therefore neglected for this set of measurements. The optical properties of apples recovered at 660 nm wavelength at SDSs of 1 and 5 mm 1, 7, and 14 days after purchase are displayed in Figure 5. The decrease over time in the $\mu_a$ and $\mu_s'$ at 660 nm indicate the breakdown of chlorophyll and the enzyme of the cell walls of the apple skin [19,20]. Figure 5a shows that the $\mu_a$ at 660 nm and 1 mm SDS decreased by 34% after the first 7 days, and 20% from Day 7 to Day 14, which indicates that the trend of breakdown of the chlorophyll

decelerates gradually. We observed that the decreasing rates of the $\mu_a$ at 1 mm SDS were more pronounced, which is 47%, than that at 5 mm SDS, which is 24%. This demonstrates that the change in chlorophyll can be detected more prominently at 1 mm. The $\mu_a$ variation at 660 nm wavelength and SDS of 5 mm was not as notable as at 1 mm, possibly because the concentration of chlorophyll is much lower in the apple outer flesh than in the skin. Figure 5b shows that the $\mu_s'$ at 660 nm wavelength at SDSs of 1 mm and 5 mm decreased in 14 days; this finding agrees with the phenomenon reported in some studies, where the decrease in the $\mu_s'$ was associated with apple softening during ripening [21]. Bobelyn et al. [22] found that the enzymatic breakdown in the cell walls leads to an increase in air-filled pores and cell size reduction. The presence of these additional air spaces also suggests the softening of the fruit.

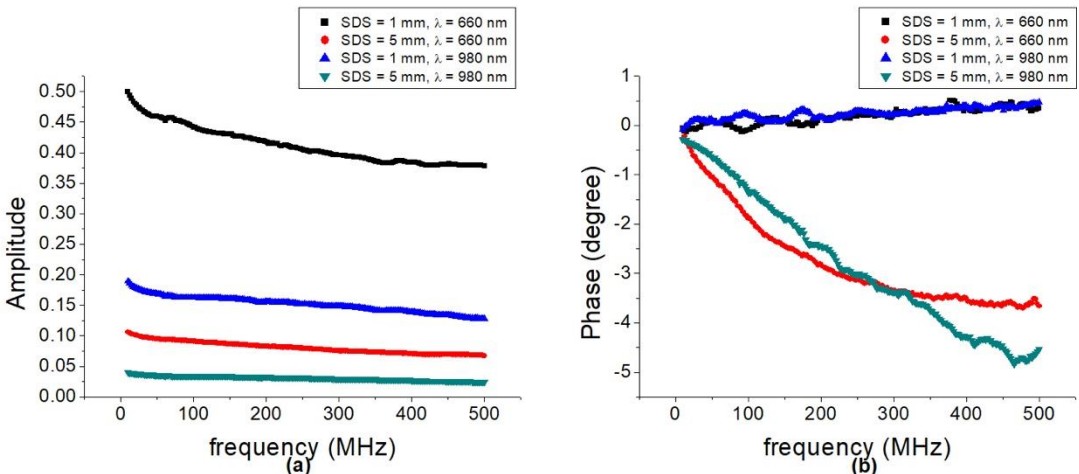

**Figure 4.** (**a**) Amplitude and (**b**) phase of apples verse the source modulation frequencies generated from different source–detector separations (SDSs) and wavelengths: 1 mm at 660 nm (■), 5 mm at 660 nm (●), 1 mm at 980 nm (▲), and 5 mm at 980 nm (▲).

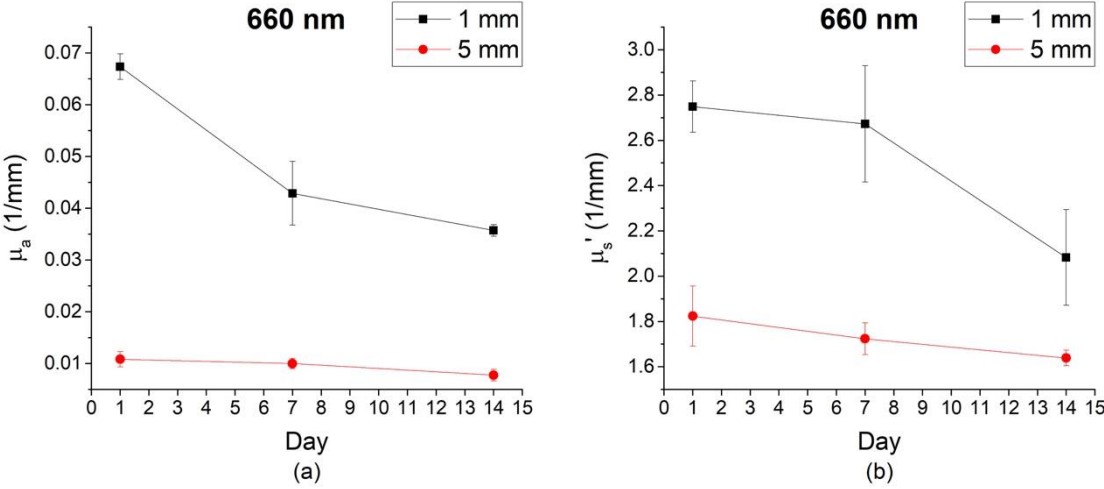

**Figure 5.** (**a**) Absorption and (**b**) reduced scattering coefficients of an apple recovered at source–detection separation (SDS) of 1 mm (black squares) and 5 mm (red circles) 1, 7, and 14 days after purchase at a wavelength of 660 nm.

In Figure 6a, the $\mu_a$ at a wavelength of 980 nm at 5 mm SDS decreased by 43% after 14 days, reflecting the reduction of the water concentration in the apple outer flesh. The rapid decrease of $\mu_a$ might be affected by the storage condition (apples were uncovered in the air-conditioned room). Moreover, it was also probably due to a coupling effect between the scattering and absorption quantification in

the frequency-domain fit. Such coupling appears when the diffusion model's assumptions are not sufficiently fulfilled [23]. The structure of apple skin and the deeper flesh tissue could be a possible source of deviation from the model. The coupling between $\mu_a$ and $\mu_s'$ caused by inaccurate modeling could be alleviated by incorporating more accurate photon transport models such as Monte Carlo method based models. Figure 6b shows the $\mu_s'$ at a wavelength of 980 nm at SDS of 5 mm also decreased in 14 days, as consistent with apple softening during ripening [21]. In Figure 6a,b, we observed the error bars of the $\mu_a$ and $\mu_s'$ at 1 mm were large; this fact suggests that the water distribution in the skin could be nonhomogeneous. In general, the water content is low in the apple skin, and we speculate that the water absorption signal may come from the superficial flesh. Thus, the uneven distribution of the skin thickness and wax of the apple skin could induce large error bars and variations in the measurement results.

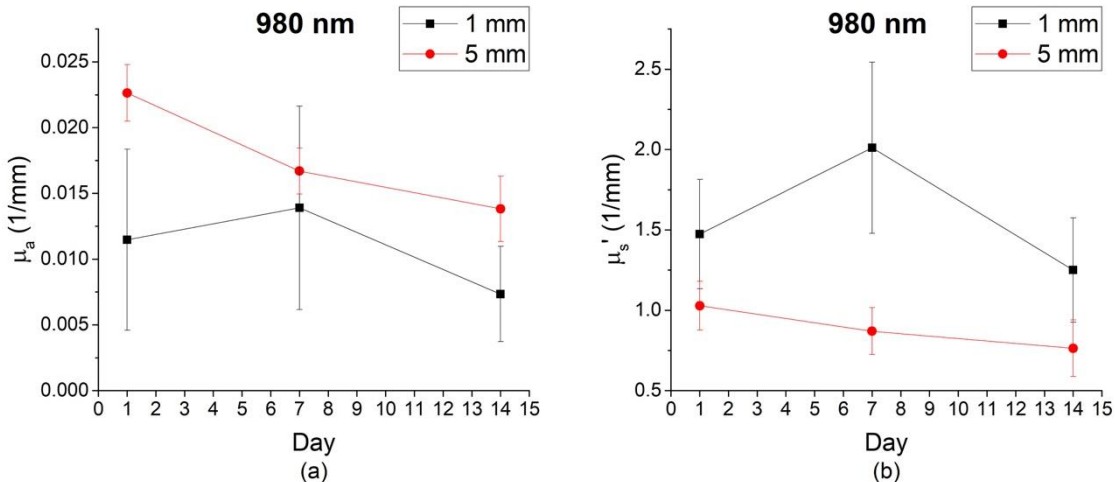

**Figure 6.** (**a**) Absorption and (**b**) reduced scattering coefficients of an apple recovered at a wavelength of 980 nm at source–detection separation (SDS) of 1 mm (black squares) and 5 mm (red circles) 1, 7, and 14 days after purchase.

From Figures 5 and 6, we conclude that the $\mu_a$ and $\mu_s'$ detected at 5 mm provide abundant information on changes in both chlorophyll and water, while the optical properties at 1 mm clearly show the information on chlorophyll.

### 3.3. Optical Properties of Gala Apples that had Irreversible Damage during Storage in Refrigerator

Consumers generally expect that apples remain fresh for a long time when kept under atmosphere-controlled storage conditions, such as in the refrigerator [24]. It is difficult to detect bruises in apples in the early stage because the skin often obscures the appearance of the underlying damage. In our study, all six apples looked intact when we purchased them on day 1; however, one of the apples had visible damage on day 7. We kept the injured apple, and analyzed the optical properties of the normal and injured apples separately. We collected five frequency-domain reflectance at 5 mm SDS from a certain spot of each apple. For the injured apple, we concentrated on the spot of visible damage. Figures 7 and 8 show the optical properties of the normal and injured apples on different days at wavelengths of 660 and 980 nm, respectively. The average $\mu_a$ and $\mu_s'$ of normal apples and injured apple are listed in Tables 1 and 2. We observed that the $\mu_a$ and $\mu_s'$ of the injured apple significantly differed from those of the normal apples. The $\mu_a$ of the normal apples at a wavelength of 660 nm decreased with time due to chlorophyll breakdown, but the $\mu_a$ of the injured apple at 660 nm increased. This is caused by the extent of browning and the phenolic composition of the apples. The composition of the apple was modified after injury, and the polyphenols in the apples were oxidized to their corresponding quinones by polyphenol oxidase; these quinones are then polymerized with other quinones and amines to form brown pigments [25,26]. Thus, the increase in the $\mu_a$ of the injured

apple, as shown in Figure 7a, indicates the accumulation of brown pigments after injury. The trend of decrease in the $\mu_a$ of the normal apples at 660 nm is 12% when they were refrigerated, compared to 24% when stored in an air-conditioned room; the $\mu_s'$ did not vary significantly ($p > 0.05$) in 14 days. Thus, we speculate that the storage of fruits in suitably cold temperature sustains the freshness longer. This finding agrees with the phenomenon that low temperature storage has the benefit of protecting nonappearance qualities: texture, nutrition, aroma, and flavor [27].

In Figure 7, a slight difference was observed in the optical properties at 660 nm between the injured and normal apples on day 1. However, the difference was not visually perceptible on day 1. The optical properties of the injured and normal apples have significant variation trends. These findings suggest that our frequency-domain DRS system has the potential to detect injured apples in the early stage.

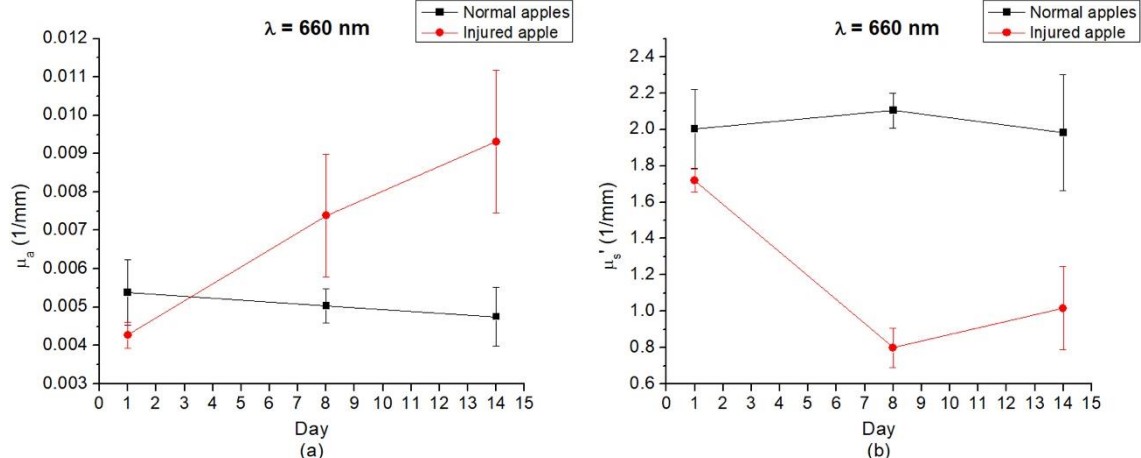

**Figure 7.** (**a**) Absorption and (**b**) reduced scattering coefficients of injured apples (red circles) and normal apples (black squares) recovered at a wavelength of 660 nm in 14 days.

**Table 1.** Optical properties of normal apples and injured apple recovered at 660 nm in 14 days.

| | Normal Apples (N = 5) | | Injured Apple (N = 1) | |
|---|---|---|---|---|
| | $\mu_a$ (mm$^{-1}$) | $\mu_s{'}$ (mm$^{-1}$) | $\mu_a$ (mm$^{-1}$) | $\mu_s{'}$ (mm$^{-1}$) |
| Day 1 | 0.0054 | 2.004 | 0.0043 | 1.718 |
| Day 7 | 0.0050 | 2.105 | 0.0074 | 0.798 |
| Day 14 | 0.0048 | 1.982 | 0.0093 | 1.017 |

Figure 8a shows that the $\mu_a$ of normal apples at 980 nm decreased slightly, but that of the injured apple increased rapidly. We suspect that the vacuolation and cytoplasmic collapse of the damaged cells caused the cell wall degradation, and caused the cytoplasm to spread out, thereby causing the increase in the $\mu_a$ of the injured apple at 980 nm. Figures 7b and 8b show that the $\mu_s'$ of the normal apples were almost invariant; however, they were significantly decreased in the injured apple in the first 7 days. We suspect that the vacuolation and cytoplasmic collapse of the damaged cells caused the $\mu_s'$ to decrease rapidly at 660 and 980 nm. From day 7 to day 14, the $\mu_s'$ of the injured apple increased. It is assumed that the extensive freezing damages caused by the change in cell structure led to the increase in the $\mu_s'$ [28].

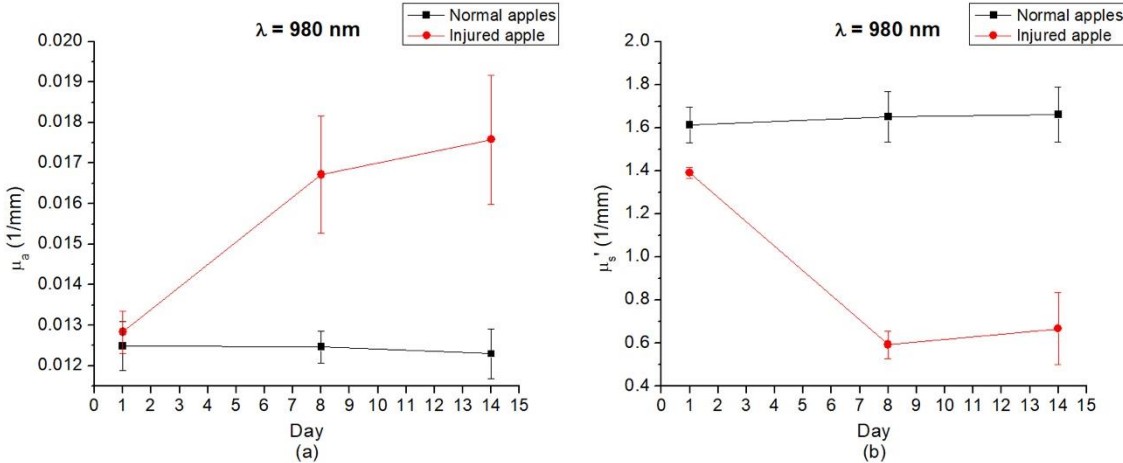

**Figure 8.** (**a**) Absorption and (**b**) reduced scattering coefficients of injured apples (red circles) and normal apples (black squares) recovered at the wavelength of 980 nm in 14 days.

**Table 2.** Optical properties of normal apples and injured apple recovered at 980 nm in 14 days.

|  | Normal Apples (N = 5) | | Injured Apple (N = 1) | |
|---|---|---|---|---|
|  | $\mu_a$ (mm$^{-1}$) | $\mu_s'$ (mm$^{-1}$) | $\mu_a$ (mm$^{-1}$) | $\mu_s'$ (mm$^{-1}$) |
| Day 1 | 0.0125 | 1.613 | 0.0128 | 1.390 |
| Day 7 | 0.0125 | 1.650 | 0.0167 | 0.592 |
| Day 14 | 0.0122 | 1.661 | 0.0176 | 0.667 |

In general, our DRS system is a potential tool for investigating the changes in the optical properties of apples. However, this study has some limitations. Firstly, the Gala apples are multi-colored fruits with high percentage of yellow coloration in addition to red pigmentation; therefore, the absorption of chlorophyll could vary greatly in different apples. Secondly, we were unable to find out the postharvest duration when we purchased the apples from the market. Therefore, the initial $\mu_a$ and $\mu_s'$ of the chlorophyll and water contents of the two groups are significantly different; this finding agrees with the phenomenon reported in other study [29]. Overall, our preliminary data shows that our DRS system provides reliable measurements of the changes in the optical properties of apples over time. In addition, we found that our DRS system could be a useful tool for screening injured apples that the damage was obscured by the skin.

## 4. Conclusions

In this study, the optical properties of apples were noninvasively determined using the proposed frequency-domain DRS system. In this system, we used two SDSs, 1 and 5 mm, and two laser diode wavelengths of 660 and 980 nm, corresponding to the chlorophyll and water absorption peaks. We found that the optical properties recovered by our DRS almost coincided with the apples' physiological changes during the 14-day study. The decrease in the $\mu_a$ and $\mu_s'$ at the wavelength of 660 nm at 1 mm SDS corresponded to the breakdown in the chlorophyll and enzyme of the cell walls. The decrease in the $\mu_a$ recovered at SDS 5 mm at the wavelength of 980 nm after 14 days suggested that the water concentration in the apple outer flesh reduced. The decrease in the $\mu_s'$ at an SDS of 5 mm and wavelength of 980 nm after 14 days were associated with apple softening during ripening. These optical properties changed more significantly with the physiological phenomena that occurred during the measurement of injured apples. Our frequency-domain DRS system could be applied to different sampling depths, especially in thin-skinned fruits, and could be used to study thick-skinned fruits by increasing the SDS. The DRS techniques might be practical tools for grading fruits or monitoring fruit changes during long-term storage.

**Author Contributions:** All authors conceived and designed the experiments. N.-Y.C. and C.-C.C. performed the analysis experiments. N.-Y.C. was responsible for manuscript writing. All Authors contributed to the interpretation and discussion of experimental results.

**Funding:** This research is supported by the Ministry of Science and Technology of Taiwan under Grant No. NSC-101-2221-E-006-211- and MOST-107-2221-E-006-148-.

**Acknowledgments:** We thank Ai-Hua, Yang from Tainan District Agricultural Research and Extension Station for valuable physiological information and suggestion of the research.

**Conflicts of Interest:** The authors declare no conflict of interest.

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
