# Peer review of "Nondestructive Evaluation of Apple Fruit Quality by Frequency-Domain Diffuse Reflectance Spectroscopy: Variations in Apple Skin and Flesh"

_applsci, doi:10.3390/app9112355_

Round 1
Reviewer 1 Report
Please see my comments in the attached pdf file.

Author Response
We would like to thank the reviewer for the useful comments and suggestions to improve the paper. We have addressed all the comments as follows. The revised sentences are highlighted in the new manuscript.

Reviewer 2 Report
The manuscript has the weaknesses that the authors explain in lines 225-231. It is unclear where the ANNs were applied, and equations (1) and (2) have problems displaying symbols. However, it is interesting to analyze the optical properties of the products to understand their response when applying spectral techniques.
Author Response

(The authors gave the same response as above.)

Reviewer 3 Report
Paper titled “Nondestructive evaluation of apple fruit quality by frequency-domain diffuse reflectance spectroscopy: variations in apple skin and flesh” is a report of utilizing DRS technique to provide insight for online fruit analysis by optical interrogation.
Some comments and question are:
1. Why do you need to sweep the frequency only in 10-500 MHz? Is this trying to achieve lock-in amplification to overcome the noise issue or other reasons?
2. Where does equation 1 and 2 which seems to be the core part of this paper originate from? Was it derived by the authors or from some reference?
3. Some sort of raw data of comparing amp and phase of apples of two cases (good and bruised) will be helpful for visualization.
4. Current output is all reported in terms of reduced scattering coefficient and absorption coefficient. What is the relationship of these optical parameters to the fruit quality? Some sort of reference or equation (if there is any) will be helpful to assess the meaningfulness of this report.
Author Response

(The authors gave the same response as above.)

Round 2
Reviewer 1 Report
Dear authors,
Thank you for the thorough revision and clarifications provided. I am satisfied with your answers and now recommend the manuscript for publication. Just one hint about the decrease of mua at 980 nm: maybe the determination of mua depends on mus', that is: although it seems that mua and mus' are independent fit parameters, there is an indirect influence of mus' on the determination of mua. There are some discussions in the literature about it. I suggest you try to look upon the references and see if the discussion applies to your setup.
Best wishes
Author Response
We deeply appreciate the reviewer’s suggestion of the reason of the decrease of mua at 980 nm. This is indeed discussed in the previous literatures, and we neglected the factors that mua will be affected by mus'. We have added the following sentences in our manuscript:
The rapid decrease of mua might be affected by the storage condition (apples were uncovered in the air-conditioned room). Moreover, it was also probably due to a coupling effect between the scattering and absorption quantification in the frequency-domain fit. Such coupling appears when the diffusion model’s assumptions are not sufficiently fulfilled [23]. The structure of apple skin and the deeper flesh tissue could be a possible source of deviation from the model. The coupling between scattering and absorption coefficients caused by inaccurate modeling could be alleviated by incorporating more accurate photon transport models such as Monte Carlo method based models.
This will be great help to our future research. Many thanks again!
Reviewer 3 Report
newer version read better and has better quality for readership
Author Response
We thank for the reviewer's help in the manuscript.